# BI 1015550 Improves Silica-Induced Silicosis and LPS-Induced Acute Lung Injury in Mice

**DOI:** 10.3390/molecules30061311

**Published:** 2025-03-14

**Authors:** Yuming Liu, Jing Zhang, Yayue Hu, Zhigang Liu, Zhongyi Yang, Ran Jiao, Xueze Liu, Xiaohe Li, Feng Sang

**Affiliations:** 1State Key Laboratory of Medicinal Chemical Biology, College of Pharmacy, Nankai University, Tianjin 300353, China; 2120231738@mail.nankai.edu.cn (Y.L.); yyhu1999@163.com (Y.H.); 1120220741@mail.nankai.edu.cn (Z.L.); 2120231679@mail.nankai.edu.cn (Z.Y.); 2120231714@mail.nankai.edu.cn (R.J.); liuxz0328@163.com (X.L.); 2School of Life Sciences and Medicine, Shandong University of Technology, Zibo 255049, China; zhangjing8947@163.com; 3Tianjin Key Laboratory of Molecular Drug Research, Tianjin International Joint Academy of Biomedicine, Tianjin 300457, China

**Keywords:** BI 1015550, PDE4B, cAMP, silicosis, acute lung injury

## Abstract

Silicosis is an interstitial lung disease (ILD) caused by prolonged inhalation of silica particles. Acute lung injury (ALI) is a critical clinical syndrome involving bilateral lung infiltration and acute hypoxic respiratory failure. However, there is currently no effective treatment for these two diseases. Previous research has established that cyclic adenosine monophosphate (cAMP) is pivotal in the pathogenesis of silicosis and acute lung injury. Phosphodiesterase 4 (PDE4) is a hydrolase enzyme of cAMP, and BI 1015550, as an inhibitor of PDE4B, is expected to be a candidate drug for treating both. BI 1015550 has shown certain anti-inflammatory and anti-fibrotic properties in systemic sclerosis-associated interstitial lung disease (SSc-ILD) and idiopathic pulmonary fibrosis (IPF), but there is a lack of research on silicosis and acute lung injury. In this research, we successfully synthesized BI 1015550 autonomously and demonstrated that it could significantly improve lung fibrosis and inflammation in a silica-induced silicosis mouse model. Furthermore, we found that BI 1015550 could also alleviate lung inflammation in a Lipopolysaccharide (LPS)-induced acute lung injury mouse model. The mechanism of action may involve the regulation of cAMP-related signaling pathways.

## 1. Introduction

Silicosis is an interstitial lung disease (ILD) caused by prolonged inhalation of silica particles [1,2]. It is characterized by inflammation, the formation of silicotic nodules, and fibrosis [3,4]. Silicosis is one of the most dangerous occupational diseases, especially in developing countries [5,6]. However, apart from the herbal alkaloid tetrandrine, there is no effective treatment for silicosis at present [7,8,9]. Therefore, there is a pressing need to develop more advanced and efficacious drugs for the treatment of silicosis.

Acute lung injury (ALI) is a critical clinical syndrome involving bilateral lung infiltration and acute hypoxic respiratory failure [10,11]. It is characterized by acute respiratory insufficiency, rapid breathing, cyanosis that does not respond to oxygen therapy, reduced lung compliance, and widespread alveolar infiltrates visible on chest X-ray [12,13,14]. Acute lung injury carries the risk of further deteriorating into acute respiratory distress syndrome (ARDS) [15]. Both acute lung injury and acute respiratory distress syndrome are mostly associated with acute and severe inflammation in lungs [16,17]. Currently, there is a significant shortage of effective treatments for acute lung injury [18]. Consequently, there is a pressing need to conduct research and develop novel therapies for acute lung injury.

Previous research has established that cyclic adenosine monophosphate (cAMP) is pivotal in the pathogenesis of silicosis and acute lung injury [19,20,21]. Liu’s research reported that dibutyryl-cAMP blocks fibroblast differentiation through the PKA/CREB/CBP signaling pathway, alleviating pulmonary fibrosis in silicosis rats [19]. Wang’s research indicated that the combination of emodin and pseudoephedrine regulates macrophage M1/M2 polarization by modulating the VIP/cAMP/PKA pathway, thereby improving LPS-induced acute lung injury [20]. cAMP plays an important regulator for cellular activities. Phosphodiesterase 4 (PDE4) has been implicated in cAMP hydrolysis in multiple inflammatory cells. Therefore, inhibition of PDE4 can suppress both immune and inflammatory cells [22,23,24,25].

BI 1015550, a phosphodiesterase 4B (PDE4B) inhibitor developed by Boehringer Ingelheim, is currently undergoing clinical trials for idiopathic pulmonary fibrosis (IPF) and progressive pulmonary fibrosis (PPF) [26]. Preclinical studies have shown that BI 1015550 possesses anti-inflammatory and anti-fibrotic characteristics in pulmonary fibrosis [27,28]. A previous study has reported that BI 1015550 could modulate the TGF-β1 pathway, which in turn ameliorates bleomycin-induced systemic sclerosis-associated interstitial lung disease in mice [29]. However, the antifibrotic properties in silicosis and the anti-inflammatory properties in acute lung injury of BI 1015550 have not been clinically verified yet. In this study, we explored a synthetic route for BI 1015550 and investigated the efficacy of BI 1015550 treatment in a silica-induced silicosis model and an LPS-induced acute lung injury model.

## 2. Results

### 2.1. BI 1015550 Alleviated Pulmonary Fibrosis in Silica-Induced Silicosis Mouse Models

Initially, our objective was to examine the mitigating impact of BI 1015550 on pulmonary fibrosis in silica-induced silicosis mouse models. The silicosis model was established by instilling 20 µL of a 400 mg/mL silica suspension intratracheally (8 mg/mouse) into mice that had been anesthetized with 1% pentobarbital sodium. Control group mice were treated with an equal volume of PBS via intratracheal administration. Ten days following silica exposure, mice were administered Nintedanib (100 mg/kg, qd) or BI 1015550 (0.75 mg/kg, bid) via daily oral gavage for a consecutive period of 20 days. We evaluated the pulmonary function of mice, and the results indicated that BI 1015550 significantly enhanced the forced vital capacity (FVC) and dynamic compliance (Cdyn), while reducing the inspiratory airway resistance (RL) and expiratory airway resistance (RE) (Figure 1A–D). Further studies indicated that BI 1015550 had the potential to lower the increased levels of hydroxyproline found in the lung tissues of mice affected by silicosis due to silica exposure (Figure 1E). Subsequently, following the instructions provided by the kit, we stained the tissue sections with hematoxylin and eosin (H&E), Masson’s trichrome, and Sirius red. The results showed that BI 1015550 significantly inhibited the infiltration of inflammatory cells in lung tissues and effectively reduced collagen deposition (Figure 1F–I). Additionally, the mRNA expression levels of Fn, Col-1, and Acta2 in the lung tissues were reduced following treatment with BI 1015550 (Figure 1J–L). In conclusion, we demonstrated that pulmonary fibrosis in mice with silica-induced silicosis was alleviated following treatment with BI 1015550.

### 2.2. BI 1015550 Alleviated Pulmonary Inflammation in Silica-Induced Silicosis Mouse Models

Having verified the substantial therapeutic benefits of BI 1015550 for lung fibrosis in mice with silica-induced silicosis, we evaluated the levels of inflammatory markers in the treated mice. We collected bronchoalveolar lavage fluid (BALF) from mice and performed cell counting, smearing, and H&E staining, and used ELISA to detect the levels of inflammatory factors and the BCA kit to measure protein concentration. Upon observation, it was noted that BI 1015550 decreased the overall cell count in BALF as well as the quantity of three types of inflammatory cells: macrophages, lymphocytes, and neutrophils (Figure 2A–E). Moreover, BI 1015550 decreased the protein concentration in BALF (Figure 2F). Subsequently, we discovered that the accumulation of inflammatory cytokines, including tumor necrosis factor-alpha (TNF-α), interleukin-1β (IL-1β), and interleukin-6 (IL-6), was significantly reduced following treatment with BI 1015550 compared to the model group (Figure 2G–I). To delve deeper into the progression of pulmonary inflammation, we utilized qRT-PCR to evaluate the mRNA expression levels of inflammatory cytokines, including *Tnf-α*, *Il-1β*, and *Il-6*, within the mice’s lung tissues. The results indicated that the model group of mice exhibited higher levels of these markers compared to the control group, and this increase was notably inhibited after the administration of BI 1015550 (Figure 2J–L). In summary, we found that BI 1015550 alleviated pulmonary inflammation in silica-induced silicosis mouse models.

### 2.3. BI 1015550 Alleviated Pulmonary Inflammation in LPS-Induced Acute Lung Injury Mouse Models

Having verified the substantial therapeutic benefits of BI 1015550 for lung fibrosis and inflammation in silica-induced silicosis mouse models, our subsequent objective was to explore the alleviating impact of BI 1015550 on acute lung injury caused by LPS in mice. Mice were given a sedative using a 1% solution of pentobarbital sodium prior to the administration of an intratracheal injection of 3 mg/kg LPS. Control group mice were treated with an equal volume of saline via intratracheal administration. The day before exposure to LPS, mice were administered Dexamethasone (10 mg/kg, qd) or BI 1015550 (0.75 mg/kg, bid) via daily oral gavage for two consecutive days. Histopathological evaluations using H&E staining demonstrated significant improvements in inflammatory cell infiltration of the lung caused by LPS in the BI 1015550-treated group compared to the model group (Figure 3A,B). Furthermore, BI 1015550 decreased the mRNA expression levels of *Tnf-α*, *Il-1β*, and *Il-6* in the lung tissues (Figure 3C–E). The results of the total cell enumeration and cellular smears from BALF indicated that BI 1015550 reduced the total cell count in BALF and the counts of three inflammatory cell types: macrophages, lymphocytes, and neutrophils (Figure 4A–E). Subsequently, we found that BI 1015550 reduced the protein concentration in BALF (Figure 4F). Specifically, we observed that the accumulation of TNF-α, IL-1β, and IL-6 in BALF was significantly reduced following treatment with BI 1015550 compared to the model group (Figure 4G–I). In conclusion, we discovered that the pulmonary inflammation in mice with LPS-induced acute lung injury was alleviated after they were treated with BI 1015550.

## 3. Discussion

Silicosis is an interstitial lung disease (ILD) caused by prolonged inhalation of silica particles [1,2]. It is characterized by inflammation, formation of silicotic nodules, and fibrosis [3,4]. It has been reported that the levels of macrophages, neutrophils, and lymphocytes, as well as those of pro-inflammatory cytokines, are significantly elevated in BALF in individuals with silicosis and in rodent models [2]. We indicated that BI 1015550 significantly enhanced the forced vital capacity (FVC) and dynamic compliance (Cdyn), while reducing the inspiratory airway resistance (RL) and expiratory airway resistance (RE), and reduced the elevated hydroxyproline content in the lung tissues. And from the perspective of Cdyn and RE, the efficacy of BI 1015550 was slightly higher than that of Nintedanib. Moreover, BI 1015550 significantly improved fibrotic pathological changes, inhibited inflammatory cell infiltration, reduced collagen deposition, and decreased the mRNA expression levels of Fn Col-1 and Acta2 in the lung tissues. Compared to Nintedanib, BI 1015550 significantly reduced the expression levels of Fn and Col-1 mRNA in the lung tissues. In conclusion, pulmonary fibrosis in mice with silica-induced silicosis was alleviated following treatment with BI 1015550. To some extent, its therapeutic effect was slightly higher than that of Nintedanib, indicating that it has the potential to become a more effective antifibrotic drug.

Having verified the substantial therapeutic benefits of BI 1015550 for lung fibrosis in mice with silica-induced silicosis, we evaluated the levels of inflammatory markers in the treated mice. BI 1015550 reduced the overall cell count in BALF, the quantities of three types of inflammatory cells—macrophages, lymphocytes, and neutrophils—and the protein concentration in BALF. It also inhibited the accumulation of inflammatory cytokines in BALF and the mRNA expression levels of *Tnf-α*, *Il-1β*, and *Il-6* in mice lung tissues. Especially in terms of the IL-1β indicator, BI 1015550’s inhibitory effect on IL-1β was slightly stronger than that of Nintedanib, whether in BALF or in lung tissue. In summary, we found that BI 1015550 alleviated pulmonary inflammation in silica-induced silicosis mouse models and that its anti-inflammatory effect was slightly stronger than that of Nintedanib, demonstrating certain advantages in terms of anti-inflammatory aspects.

Research indicates that cAMP signaling could possess anti-silicotic properties [30]. Upon binding of a particular ligand to a G protein-coupled receptor, adenylyl cyclase becomes activated, which in turn causes an increase in intracellular cAMP levels. Subsequently, this triggers the activation of downstream molecules, such as protein kinase A and EPAC [29]. Rises in cAMP suppress the multiplication of fibroblasts and the creation of the extracellular matrix (ECM), providing anti-fibrotic benefits both in vivo and in vitro [31]. Therefore, BI 1015550, as a PDE4B inhibitor, may exert its therapeutic effect by increasing cAMP levels in silicosis, and the same mechanism of action of BI 1015550 has been confirmed in SSc-ILD. Furthermore, we found that the anti-inflammatory and anti-fibrotic effects of BI 1015550 are slightly stronger than those of Nintedanib. We believe this may be due to BI 1015550 simultaneously inhibiting the core pathways of inflammation and fibrosis (such as TGF-β, cAMP-PKA), and this dual inhibition demonstrates stronger anti-inflammatory and anti-fibrotic efficacy [29]. Additionally, Nintedanib mainly targets fibroblasts and endothelial cells, with weaker regulation during the inflammatory phase. BI 1015550, on the other hand, can act on both inflammatory cells (macrophages, neutrophils) and structural cells (fibroblasts, epithelial cells), blocking multiple stages of the fibrotic process [28]. Therefore, the anti-inflammatory and anti-fibrotic effects of BI 1015550 may be stronger than those of Nintedanib, and the specific mechanisms require further study.

Acute lung injury (ALI) is a critical clinical syndrome involving bilateral lung infiltration and acute hypoxic respiratory failure [10,11]. It is characterized by acute respiratory insufficiency, rapid breathing, cyanosis that does not respond to oxygen therapy, reduced lung compliance, and widespread alveolar infiltrates visible on chest X-ray [12,13,14]. During the initial phase of lung damage, there is a rapid increase in the levels of inflammatory mediators like TNF-α, IL-1β, and IL-8 within the bronchoalveolar lavage fluid (BALF). The equilibrium of these inflammatory agents fosters pulmonary inflammation [32]. Our research results demonstrated that BI 1015550 significantly improved inflammatory cell infiltration of lung caused by LPS, and reduced the mRNA expression levels of *Tnf-α*, *Il-1β*, and *Il-6* in the lung tissues. The results of the total cell enumeration and cellular smears from BALF indicated that BI 1015550 reduced the total cell count in BALF, as well as the counts of three inflammatory cell types: macrophages, lymphocytes, and neutrophils. It also inhibited the protein concentration in BALF and the accumulation of inflammatory cytokines. However, considering all the experimental results, the anti-inflammatory effect of BI 1015550 is weaker than that of Dexamethasone. In summary, we discovered that pulmonary inflammation in mice with LPS-induced acute lung injury was alleviated after being treated with BI 1015550.

An increasing amount of evidence suggests that regulating the cAMP pathway can effectively improve LPS-induced ALI. Pseudoephedrine and emodin treatment has been shown to enhance the expression of VIP cAMP, as well as that of p-PKA protein in lung tissues, and to markedly suppress the phosphorylation of NF-κB and MAPK. Furthermore, this therapeutic approach can prevent M1 polarization and encourage M2 polarization by utilizing the VIP/cAMP/PKA signaling pathway [20]. The cAMP-PKA pathway acts to suppress the activation of p38 MAPK, ERK1/2, and JNK, causing a reduction in AP-1- and C/EBP-mediated transcription, which helps to alleviate IgG-IC-associated acute lung injury [21]. Similarly, BI 1015550, as an inhibitor of PDE4B, may include regulating cAMP as its mechanism of action. As for why the anti-inflammatory effect of BI 1015550 is weaker than that of Dexamethasone, we believe it may be because Dexamethasone, as a glucocorticoid, has a very broad range of effects capable of influencing gene transcription levels and simultaneously inhibiting multiple inflammatory pathways and mediators. In contrast, BI 1015550, as a PDE4B inhibitor, has a relatively more limited mechanism of action. Since PDE4B is primarily found in inflammatory cells, although inhibiting PDE4B may lead to an increase in intracellular cAMP levels, thereby suppressing the activation of these cells and the release of inflammatory mediators, this effect may be limited to specific inflammatory cells and mediators, unlike glucocorticoids, which affect a wide range of inflammatory pathways. Therefore, its anti-inflammatory effect is not as strong as that of Dexamethasone. However, considering safety, long-term use of Dexamethasone may lead to severe side effects (such as immunosuppression, metabolic disorders); thus, BI 1015550 may serve as a milder anti-inflammatory drug that can be used for long-term treatment to inhibit disease progression. The specific mechanism of BI 1015550 in treating ALI requires further study.

## 4. Experimental Materials and Methods

### 4.1. Materials

The preparation of BI 1015550 referred to the synthesis method reported by Pouzet P. et al. [33] (Appendix A). The HRMS data and NMR data of BI 1015550 are included in the Appendix A).

### 4.2. Laboratory Animals

C57BL/6J male mice, aged 6–8 weeks, were supplied by Beijing Vital River Laboratory Animal Technology Co., Ltd. (Beijing, China). The care of the animals and the experimental procedures were authorized by the Experimental Animal Care and Use Committee of Nankai Animal Resources Center, under approval number 2024-SYDWLL-000001. All procedures were carried out in compliance with the pertinent guidelines and regulations.

### 4.3. Silicosis Model

Crystalline silica was obtained from HWRK Chemical Co., Ltd. (CAS14808-60-7; purity 99.9%; Beijing, China). Silica particles were baked at 180 °C for at least 2 h; then, they were suspended in sterile phosphate-buffered saline (PBS) and sonicated for 20 min.

The silicosis model was established by instilling 20 µL of a 400 mg/mL silica suspension intratracheally (8 mg/mouse) into mice that had been anesthetized with 1% pentobarbital sodium. Control group mice were treated with an equal volume of PBS via intratracheal administration. Nintedanib was employed as the positive control. Ten days following silica exposure, mice were administered Nintedanib (100 mg/kg, once daily) or BI 1015550 (0.75 mg/kg, twice daily) via daily oral gavage for a consecutive period of 20 days. On the 30th day, mice from all experimental groups were anesthetized to evaluate their pulmonary function, followed by the collection of bronchoalveolar lavage fluid (BALF) for counting, smearing, and an enzyme-linked immunosorbent assay (ELISA). Subsequently, the right lung was collected for quantitative reverse-transcription PCR (qRT-PCR) and hydroxyproline content testing, while the left lung was collected for histopathological testing (Figure 5A).

A total of 20 mice were randomly divided into four groups, with 5 mice per group. These groups were designated as follows: the PBS + Vehicle group (control), the Si + Vehicle group (model), the Si + Nintedanib (100 mg/kg, qd) group, and the Si + BI 1015550 (0.75 mg/kg, bid) group.

### 4.4. Acute Lung Injury Model

Mice were given a sedative using a 1% solution of pentobarbital sodium prior to the administration of an intratracheal injection of 3 mg/kg LPS (Sigma, St. Louis, MO, USA, L2880-10 mg). Control group mice were treated with an equal volume of saline via intratracheal administration. Dexamethasone was used as the positive control. The day before exposure to LPS, mice were administered Dexamethasone (10 mg/kg, once daily) or BI 1015550 (0.75 mg/kg, twice daily) via daily oral gavage for two consecutive days. Next, the mice were sacrificed, and bronchoalveolar lavage fluid (BALF) was collected for cell counting, smearing, and an enzyme-linked immunosorbent assay (ELISA). Subsequently, the right lung was collected for quantitative reverse-transcription PCR (qRT-PCR), and the left lung was collected for histopathological testing (Figure 5B).

A total of 20 mice were randomly divided into four groups, with 5 mice per group. These groups were designated as follows: the control group, the LPS group (model), the LPS + Dexamethasone (10 mg/kg, qd) group, and the LPS + BI 1015550 (0.75 mg/kg, bid) group.

### 4.5. Tests of Lung Function

The pulmonary function of mice were evaluated using the AniRes2005 system for animal pulmonary function analysis, supplied by Beijing BoiLab Co., Ltd. (Beijing, China). After the mice were anesthetized, their head and limbs were quickly secured. A syringe sleeve was used to elevate the neck to facilitate intubation and prevent airway obstruction that could lead to asphyxiation. The skin was quickly cut open along the sternum towards the head, and the trachea and jugular vein were bluntly separated. A small incision was made on the trachea near the head, and the tracheal connector was inserted into the trachea towards the lungs. The tracheal connector was quickly double-knotted at both ends of the intubation site to secure it in the trachea. The mice were then quickly transferred to the body-scanning platform, and the ventilator was connected to the tracheal connector to provide passive breathing support for the mice. The ventilator was connected to the data processing system, using pulmonary function analysis software to display various lung function parameters in real time, including forced vital capacity (FVC), dynamic compliance (Cdyn), inspiratory airway resistance (RL), and expiratory airway resistance (RE).

### 4.6. Hydroxyproline Level Testing

Lung tissues from mice were placed in vials and dried at 120 °C for 18 h. Subsequently, 3 mL of 6 M hydrochloric acid was added to each vial, which was then sealed with plastic wrap, capped tightly, and heated in a 120 °C oven for another 18 h. Once the tissue was completely dissolved, the solution was filtered and transferred to a new tube. The hydrolysis reaction was terminated by adding 6 M NaOH solution to each tube, and the pH value was adjusted to between 6.5 and 8.0. The solution was then brought up to 10 mL with PBS and shaken. Several tubes were prepared, to which 350 μL of H_2_O and 50 μL of sample were added. Concurrently, the standard was gradient-diluted and 400 μL was placed in tubes. Then, 200 μL of chloramine-T was added to each tube, mixed by vortex, and left at room temperature for 20 min, thereby oxidizing the free hydroxyproline to pyrrole. Following this, 200 μL of perchloric acid was added to each tube, mixed by vortex, and left at room temperature for 5 min to terminate the oxidation reaction. Subsequently, 200 μL of P-DMAB was added to each tube, mixed by vortex, and left in a 50 °C water bath for 20 min to allow P-DMAB to fully react with pyrrole to generate a red compound. After cooling, 200 μL from each tube was transferred to a 96-well plate, and the absorbance was measured at 577 nm. The standard curve was established using the standard concentration and absorbance, after which the absorbance was brought into the standard curve to measure the hydroxyproline content of each sample.

### 4.7. Organization Pathology Testing

The left lung tissues were collected and fixed in 4% paraformaldehyde for 48 h. Subsequently, they were dehydrated using a dehydrator (Leica Biosystems, Nußloch, Germany) and embedded in paraffin with an embedding machine (Leica Biosystems, Germany). The wax pieces were then cut into 5 μm thick sections using a microtome (Leica Biosystems, Germany) and placed in an oven for 3 h. The paraffin was removed using xylene, and the sections were rehydrated through a series of alcohols. Subsequently, they were stained with hematoxylin and eosin (H&E) (Solarbio, Beijing, China, G1120), Masson’s trichrome stain (Solarbio, G1340), and Sirius red stain (Solarbio, G1473), following the instructions provided in the kit. Random images were captured using a Nikon pathology microscope (Tokyo, Japan). The total area of lung tissue and the area of the fibrotic region in lung tissue were measured using Image-Pro Plus 6.0, and then the degree of lung fibrosis in each image was calculated using the following formula: fibrotic area/total area of lung tissue. The degree of lung inflammation was assessed by Szapiel, with scores ranging from 0 (no inflammation), 1 (mild inflammation), and 2 (moderate inflammation) to 3 (severe inflammation). The area of lung tissue and collagen tissue regions (blue regions stained with Masson’s trichrome or red regions stained with Sirius red) was calculated using Image-Pro Plus 6.0. Then, the collagen volume fraction of lung tissue in each image was calculated using the following formula: collagen tissue area/total area of lung tissue.

### 4.8. Quantitative Real-Time Polymerase Chain Reaction (qRT-PCR)

Lung tissue was collected in tubes and placed on ice. Then, 1 mL of Trizol was added, and the tissue was rapidly homogenized and sonicated before being left to stand for 5 min. To each sample, 200 μL of chloroform was added, mixed by vortex, and allowed to sit at room temperature for 3 min, followed by centrifugation at 12,000 rpm for 15 min at 4 °C. The supernatant was removed, and an equal volume of isopropanol was added, mixed, and left at room temperature for 10 min, followed by centrifugation at 12,000 rpm for 10 min at 4 °C. The supernatant was discarded, and the precipitate was washed with 1 mL of 75% ethanol (containing DEPC water) and centrifuged at 7000 rpm for 5 min at 4 °C. Washing was repeated once, the supernatant was discarded, and the precipitate was left until the ethanol was completely volatilized. Then, 15 μL of DEPC water was added to dissolve the RNA, and the solution was left on ice for 15 min. The RNA concentration was determined using an ultramicro spectrophotometer (DeNovix, Wilmington, DE, USA), and all RNA samples were diluted to the same concentration with DEPC water. The RNA was then reverse-transcribed to cDNA using a reverse transcription kit (TIANGEN, Beijing, China). Quantitative real-time PCR (qRT-PCR) was conducted with the Hieff UNICON^®^ qPCR SYBR Green Master Mix (YEASEN, Shanghai, China). Sequences of specific primer sets are shown in Table 1. All qRT-PCR results were normalized using the endogenous gene GAPDH as a reference. Gene expression was quantified using the 2^−∆∆CT^ method, with GAPDH serving as the reference. Here, ∆CT represents the difference between the CT values of the target gene and the reference gene, while ∆∆CT represents the difference between the ∆CT values of the treatment group and the control group.

### 4.9. Bronchoalveolar Lavage Fluid (BALF) Collection

Once the mice were anesthetized, the trachea was isolated and a retained needle was inserted into the trachea. The chest cavity was then opened, and a 1 mL syringe was utilized to aspirate pre-chilled PBS solution into the lungs. The lungs were thoroughly rinsed 1–2 times to collect BALF. The collected BALF was centrifuged for 15 min at 3000 rpm. An amount of 5 μL of the supernatant was measured for the protein concentration using the BCA protein assay kit (TIANGEN, Beijing, China), after which the supernatant was stored at −80 °C for the ELISA detection of inflammatory factors. One milliliter of RBC lysis buffer was added to the precipitate and thoroughly mixed. The mixture was held at room temperature for 5 min, and then centrifuged again for 15 min at 3000 rpm. The supernatant was discarded, and 200 μL of PBS was added to the precipitate and mixed. A 20 μL aliquot of the cell suspension was taken to count the cell numbers using an automated cell counter. The remaining part of each suspension was used for smearing. The BALF was fixed with a 4% formaldehyde solution for 30 min after being air-dried, followed by gently rinsing off the formaldehyde with water. The slides were dried, stained with H&E, and then mounted for preservation. Different inflammatory cells, including macrophages, lymphocytes, and neutrophils, were identified using standard morphological criteria under an optical microscope.

### 4.10. Enzyme-Linked Immunosorbent Assay (ELISA)

ELISA kits specific to TNF-α (JL10484, Jianglai Bio, Shanghai, China), IL-1β (JL18442, Jianglai Bio, Shanghai, China), and IL-6 (JL20268, Jianglai Bio, Shanghai, China) were employed to measure the levels of inflammatory cytokines in the bronchoalveolar lavage fluid (BALF) supernatant, adhering to the protocols provided by the manufacturers.

### 4.11. Statistical Analysis

The data were processed with GraphPad Prism 9.5, and the results are presented as Mean ± SEM. Differences between three or more groups were analyzed by one-way analysis of variance (ANOVA) with Tukey’s post hoc multiple comparison tests. *p*-values less than 0.05 were deemed statistically significant.

## 5. Conclusions

In conclusion, we successfully synthesized BI 1015550 autonomously and demonstrated its significant potential to improve lung fibrosis and inflammation in a silica-induced silicosis mouse model. Additionally, we observed that BI 1015550 could alleviate lung inflammation in an LPS-induced acute lung injury mouse model. The mechanism of action appears to involve the regulation of cAMP-related signaling pathways, although specific details require further investigation. Our research findings suggest that BI 1015550 represents a promising approach for treating silicosis and acute lung injury. Moving forward, we plan to delve deeper into the mechanism of action of BI 1015550 within these two disease models, with the anticipation that it will present improved prospects for the treatment of silicosis and acute lung injury in the future.

## Figures and Tables

**Figure 1 molecules-30-01311-f001:**
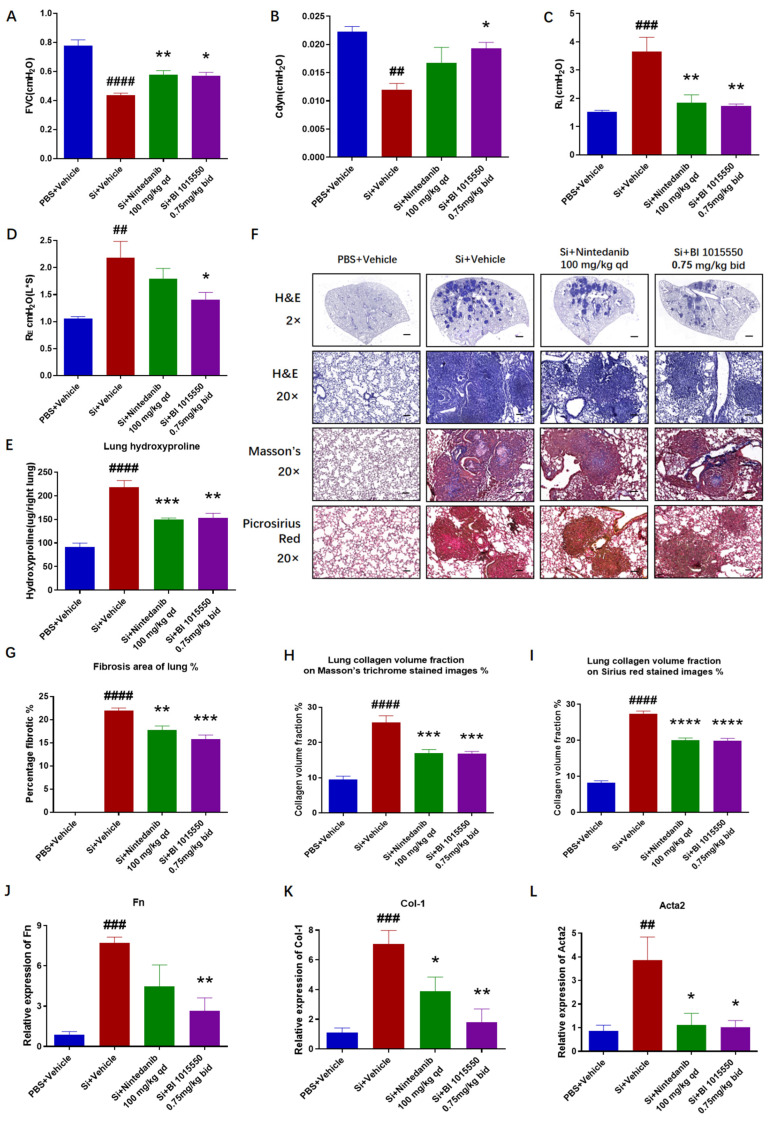
BI 1015550 significantly ameliorated pulmonary fibrosis in silica-induced silicosis mouse models. (**A**–**D**) Pulmonary function parameters, including FVC, Cdyn, RL, and RE, in each group of mice. (**E**) Hydroxyproline content in the lung tissues of each group of mice. (**F**) Representative lung sections with H&E staining (2× magnification, scale bar = 500 μm and 20× magnification, scale bar = 50 μm), Masson’s trichrome staining (20× magnification, scale bar = 50 μm), and Sirius red staining (20× magnification, scale bar = 50 μm). (**G**) Percentage of lung fibrosis area in each group of mice. (**H**) Quantification of collagen density in lung tissues on Masson’s trichrome-stained images. (**I**) Quantification of collagen density in lung tissues on Sirius red-stained images. (**J**–**L**) Expression of fibrotic markers *Fn*, *Col-1*, and *Acta2* genes in the lung tissues of mice with silicosis induced by silica. Data are presented as Mean ± SEM (one-way ANOVA with Tukey’s post hoc multiple comparison tests), *n* = 5. We use # to indicate the difference between the PBS and silica groups (^##^ *p* < 0.01; ^###^ *p* < 0.001; ^####^ *p* < 0.0001) and * to represent the difference between the silica and treatment groups (* *p* < 0.05; ** *p* < 0.01; *** *p* < 0.001; **** *p* < 0.0001).

**Figure 2 molecules-30-01311-f002:**
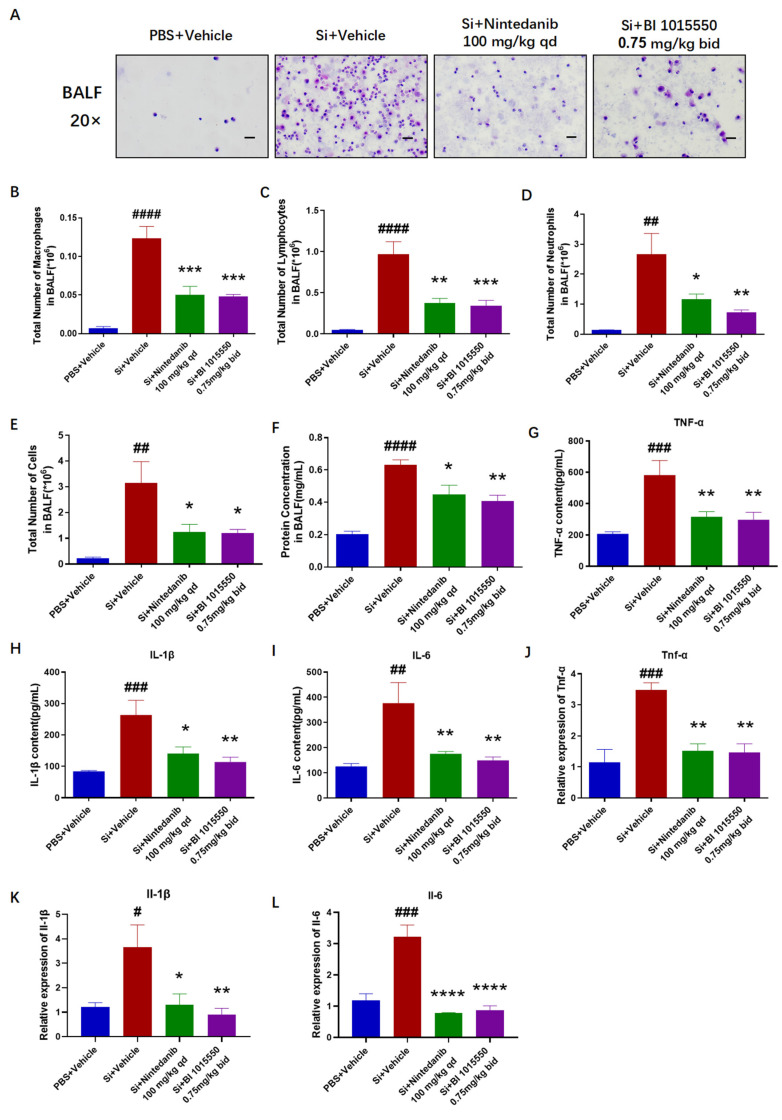
BI 1015550 mitigated pulmonary inflammation in silica-induced silicosis mouse models. (**A**) Observation of BALF smears under a microscope (20× magnification, scale bar = 50 μm). (**B**–**D**) Counts of macrophages, lymphocytes, and neutrophils in BALF after smearing and H&E staining. (**E**) Total cell count in BALF. (**F**) Results of BCA protein assay, used to determine the protein concentration in BALF. (**G**–**I**) Results of ELISA, used to detect the protein levels of inflammatory cytokines in BALF, including TNF-α, IL-1β, and IL-6. (**J**–**L**) Results of qRT-PCR, used to detect the mRNA levels of inflammatory cytokines in lung tissues, including *Tnf-α*, *Il-1β*, and *Il-6*. Data are presented as Mean ± SEM (one-way ANOVA with Tukey’s post hoc multiple comparison tests), *n* = 5. We use # to indicate the difference between the PBS and silica groups (^#^ *p* < 0.05; ^##^ *p* < 0.01; ^###^
*p* < 0.001; ^####^ *p* < 0.0001) and * to represent the difference between the silica and treatment groups (* *p* < 0.05; ** *p* < 0.01; *** *p* < 0.001; **** *p* < 0.0001).

**Figure 3 molecules-30-01311-f003:**
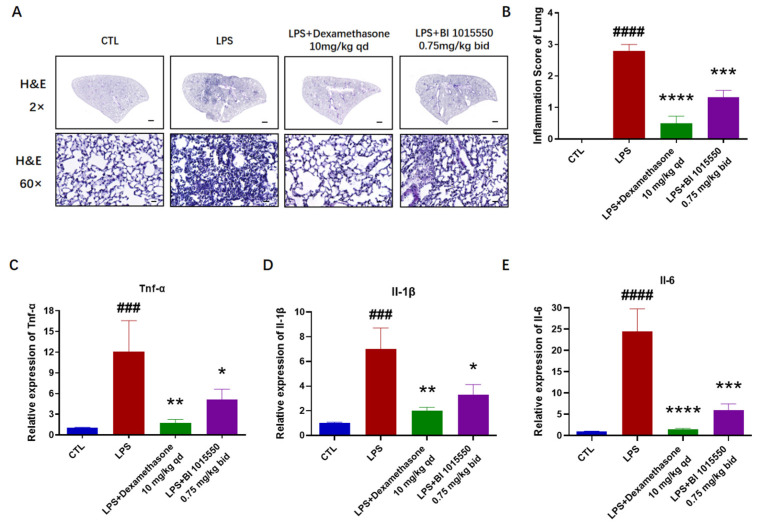
BI 1015550 alleviated pulmonary inflammation in lung tissues in LPS-induced acute lung injury mouse models. (**A**,**B**) Representative lung sections indicated by H&E staining (2× magnification, scale bar = 500 μm and 60× magnification, scale bar = 20 μm) and inflammation score of the lung. (**C**–**E**) Results of qRT-PCR, used to detect the mRNA levels of inflammatory cytokines in lung tissues, including *Tnf-α*, *Il-1β*, and *Il-6*. Data are presented as Mean ± SEM (one-way ANOVA with Tukey’s post hoc multiple comparison tests), *n* = 5. We use # to indicate the difference between the CTL and LPS groups (^###^ *p* < 0.001; ^####^ *p* < 0.0001) and * to represent the difference between the LPS and treatment groups (* *p* < 0.05; ** *p* < 0.01; *** *p* < 0.001; **** *p* < 0.0001).

**Figure 4 molecules-30-01311-f004:**
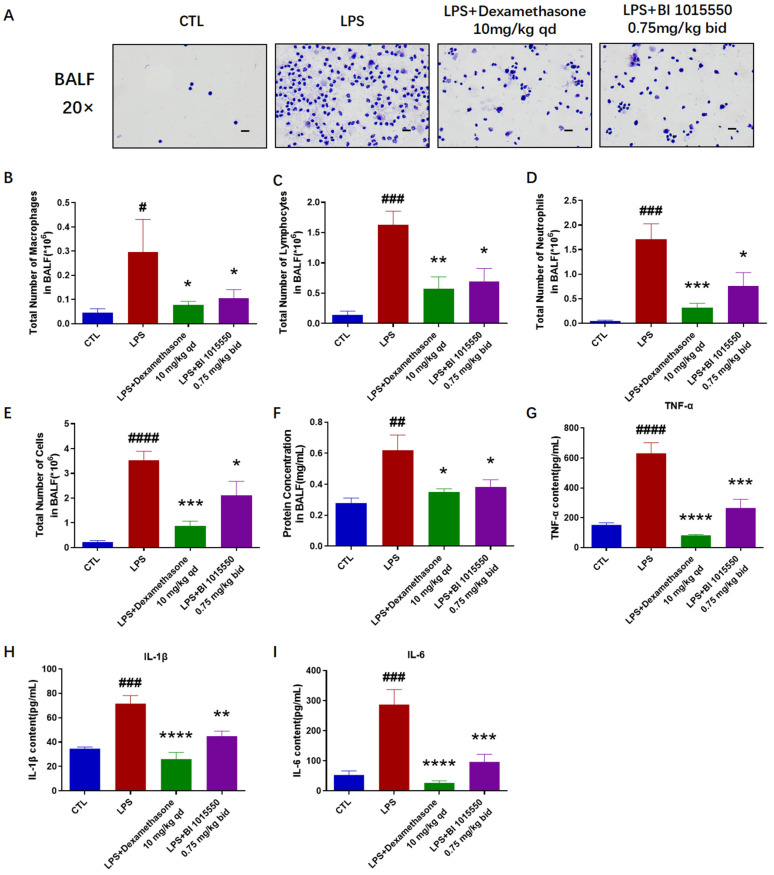
BI 1015550 alleviated pulmonary inflammation in BALF in LPS-induced acute lung injury mouse models. (**A**) Observation of BALF smears under a microscope (20× magnification, scale bar = 50 μm). (**B**–**D**) Counts of macrophages, lymphocytes, and neutrophils in BALF after smearing and H&E staining. (**E**) Total cell count in BALF. (**F**) Results of BCA protein assay, used to determine the protein concentration in BALF. (**G**–**I**) Results of ELISA, used to detect the protein levels of inflammatory cytokines in BALF, including TNF-α, IL-1β, and IL-6. Data are presented as Mean ± SEM (one-way ANOVA with Tukey’s post hoc multiple comparison tests), *n* = 5. We use # to indicate the difference between the CTL and LPS groups (^#^ *p* < 0.05; ^##^ *p* < 0.01; ^###^ *p* < 0.001; ^####^
*p* < 0.0001) and * to represent the difference between the LPS and treatment groups (* *p* < 0.05; ** *p* < 0.01; *** *p* < 0.001; **** *p* < 0.0001).

**Figure 5 molecules-30-01311-f005:**
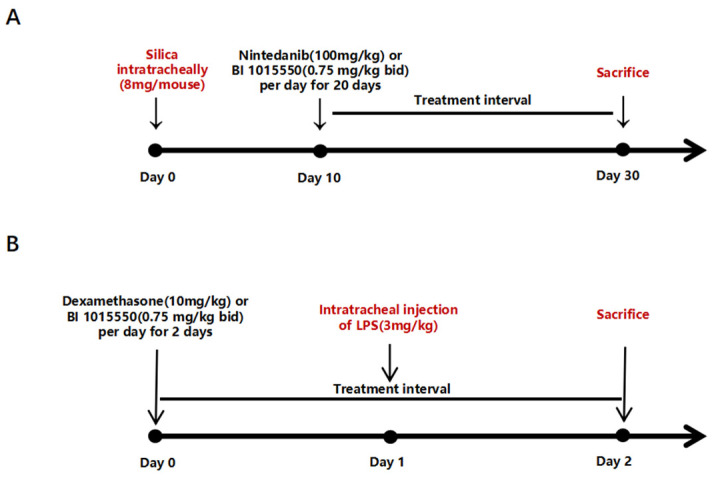
Dosing regimen in the two animal models. (**A**) Mouse silicosis models were established by the intratracheal instillation of a silica suspension (8 mg/mouse) or PBS. Ten days after exposure to silica, Nintedanib (100 mg/kg, qd) or BI 1015550 (0.75 mg/kg, bid) was administered to mice by daily oral gavage for 20 consecutive days (*n* = 5 for each group). (**B**) Mouse acute lung injury models were established by the intratracheal injection of LPS (3 mg/kg) or saline. One day before exposure to LPS, Dexamethasone (10 mg/kg, qd) or BI 1015550 (0.75 mg/kg, bid) was administered to mice by daily oral gavage for 2 consecutive days (*n* = 5 for each group).

**Table 1 molecules-30-01311-t001:** Primer sequences for qRT-PCR assay.

Gene	Primer	Sequence (5′-3′)
*Gapdh-M*	Forward Reverse	AGGTCGGTGTGAACGGATTTG GGGGTCGTTGATGGCAACA
*Fn-M*	Forward Reverse	ATGTGGACCCCTCCTGATAGT GCCCAGTGATTTCAGCAAAGG
*Collagen-I-M*	Forward Reverse	GCTCCTCTTAGGGGCCACT ATTGGGGACCCTTAGGCCAT
*Acta-2-M*	Forward Reverse	CCCAGACATCAGGGAGTAATGG TCTATCGGATACTTCAGCGTCA
*Tnf-α-M*	Forward Reverse	CAGGCGGTGCCTATGTCTC CGATCACCCCGAAGTTCAGTAG
*Il-1β-M*	Forward Reverse	GAAATGCCACCTTTTGACAGTG TGGATGCTCTCATCAGGACAG
*Il-6-M*	Forward Reverse	CTGCAAGAGACTTCCATCCAG AGTGGTATAGACAGGTCTGTTGG

## Data Availability

The original contributions presented in this study are included in the article/Appendix A. Further inquiries can be directed to the corresponding authors.

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
