# Peer review of "BI 1015550 Improves Silica-Induced Silicosis and LPS-Induced Acute Lung Injury in Mice"

_molecules, 2025, doi:10.3390/molecules30061311_

Round 1
Reviewer 1 Report
Comments and Suggestions for Authors
The manuscript titled “BI 1015550 improves silica-induced silicosis and LPS-induced acute lung injury in mice”, is a well thought out research article. Basing on the current literature, the authors tried to link anti-inflammatory and anti-fibrotic properties of BI 1015550 (an inhibitor of PDE4B), to silicosis and LPS-induced acute lung injury. Manuscript writing is excellent and satisfactory. Further I haven’t found any deviations in language or grammatical errors. However, there are couple of points which the authors need to address-
1) How did the authors arrive at the BI-1015550 in vivo dosage and concentration? Did they do a thorough study on the dose response and pharmacokinetics of the drug they used?
2) The Font within the figures is very small and too unreadable…
3) One major question is that the authors did mouse TNF-a, IL-6 and IL1b BALF cytokine estimation (Fig 2 G, H, I) and (Fig 4 G, H, I) by ELISA. However, the cytokine concentration of the above said cytokines is way below normal. Is the detection limit of these cytokines provided by the kits is way below normal?
Leaving these couple of points, the research article is written excellently and I have found no unnecessary references or self-citations and I suggest the Editors to accept the manuscript in the present form with minor modifications.
Comments on the Quality of English LanguageManuscript writing is excellent and satisfactory. Haven’t found any deviations in language or grammatical errors.
Author Response
Dear Reviewers,
Thank you for your constructive feedback. We have carefully revised the manuscript based on the comments. Below is our point-by-point response.
Comments 1: How did the authors arrive at the BI-1015550 in vivo dosage and concentration? Did they do a thorough study on the dose response and pharmacokinetics of the drug they used?
Response 1:Thank you for raising the profound question regarding the concentration selection of BI 1015550 in our experiments. The concentration of BI 1015550 was determined based on previous literature reports and our evaluation of the therapeutic effects of these compounds within a biologically relevant range.
Specifically, previous studies have shown that doses of 2.5mg/kg and 12.5mg/kg of BI1015550 (Nerandomilast) can exert anti-inflammatory and anti-fibrotic effects in a mouse model of systemic sclerosis-associated interstitial lung disease (SSc-ILD) [1]. Additionally, a literature report indicates that doses of 0.25mg/kg, 0.75mg/kg, and 2.5mg/kg of BI 1015550 can improve the levels of related inflammatory factors in the bronchoalveolar lavage fluid (BALF) of silica-induced silicosis mice to varying degrees[2]. In subsequent studies, we found that the effect of a 0.25mg/kg dose of BI 1015550 was not significant. Therefore, we chose to conduct experiments with a dose of 0.75mg/kg to ensure that our findings reflect a clear biological effect.
We appreciate your attention to detail and thank you for your valuable input.
[1]Yuming Liu, Zhigang Liu, Xiaohe Li,Wenqi Li, et al. Nerandomilast Improves Bleomycin-Induced Systemic Sclerosis-Associated Interstitial Lung Disease in Mice by Regulating the TGF-β1 Pathway. Inflammation, 2024. doi: 10.1007/s10753-024-02153-9.
[2]Herrmann FE, Hesslinger C, Wollin L, et al. BI 1015550 is a PDE4B Inhibitor and a Clinical Drug Candidate for the Oral Treatment of Idiopathic Pulmonary Fibrosis. Front Pharmacol, 2022, 13: 838449. doi: 10.3389/fphar.2023.1219760.
Comments 2: The Font within the figures is very small and too unreadable.
Response 2:We sincerely appreciate your insightful comments, which have helped us improve the quality of this manuscript. We apologize for the inconsistent font sizes in the article. We will correct the font size of the figures in the article and reformat it to make the data more clearly visible. We appreciate your valuable input.
Comments 3: One major question is that the authors did mouse TNF-a, IL-6 and IL1b BALF cytokine estimation (Fig 3 G, H, I) and (Fig 5 G, H, I) by ELISA. However, the cytokine concentration of the above said cytokines is way below normal. Is the detection limit of these cytokines provided by the kits is way below normal?
Response 3:Dear Reviewer,Thank you for your inquiry. I apologize, here we mistakenly treated the absorbance value as the concentration value for the vertical axis in data processing. We have made changes to the article (Fig 3 G, H, I) and (Fig 5 G, H, I), and if necessary, we can provide the original data for your review. Finally, we apologize once again for this oversight. We appreciate your attention to detail and thank you for your valuable input.
Sincerely,
Yuming Liu
Reviewer 2 Report
Comments and Suggestions for Authors
After reading the article BI 1015550 improves silica-induced silicosis and LPS-induced acute lung injury in mice, I found the results very interesting and the way the experiments were structured is adequate to meet the objective. However, I have some minor comments.
1. Homogenize the font within the text, since there is large and small font.
2. Could the results obtained be a consequence of pharmacokinetic changes? If so, discuss them.
3. Attach the doi in the references.
Author Response
Dear Reviewers,
Thank you for your constructive feedback. We have carefully revised the manuscript based on the comments. Below is our point-by-point response.
Comments 1: Homogenize the font within the text, since there is large and small font.
Response 1:We sincerely appreciate your insightful comments, which have helped us improve the quality of this manuscript. I apologize for the issue with inconsistent font sizes in the article. We have already adjusted the font sizes and homogenized them. Thank you for bringing these points to our attention.
Comments 2: Could the results obtained be a consequence of pharmacokinetic changes? If so, discuss them.
Response 2:Thank you for your review and feedback on our manuscript. In this study, we referenced the pharmacokinetic data of a similar drug, Roflumilast (a phosphodiesterase 4 inhibitor), which has a half-life of approximately 3-5 hours in mice after oral administration and requires twice-daily dosing. This provided guidance for our selection of the drug dosage for BI 1015550. Subsequently, we took note of an article on the efficacy of BI 1015550 published by the manufacturer, Boehringer Ingelheim, which reported that doses of 0.25mg/kg, 0.75mg/kg, and 2.5mg/kg of BI 1015550 could improve the levels of related inflammatory factors in the bronchoalveolar lavage fluid (BALF) of mice with silica-induced silicosis, all with twice-daily dosing[1]. This provided a reference for our selection of BI 10155550 to treat a mouse model of silica-induced silicosis. In subsequent research, we found that the effect of BI 1015550 at a dose of 0.25mg/kg bid was not significant. Therefore, we chose to conduct experiments with a dose of 0.75mg/kg bid to ensure that our research results reflected a clear biological effect. We appreciate your valuable input.
[1]Herrmann FE, Hesslinger C, Wollin L, et al. BI 1015550 is a PDE4B Inhibitor and a Clinical Drug Candidate for the Oral Treatment of Idiopathic Pulmonary Fibrosis. Front Pharmacol, 2022, 13: 838449. doi: 10.3389/fphar.2023.1219760.
评论3:在参考文献中附上 doi。
回复3:尊敬的审稿人, 感谢您的宝贵反馈和建议。感谢您提请我们注意这个问题,这将有助于提高手稿的严谨性。我们在参考文献的末尾附上了相应的 doi(第 471-541 行)。感谢您对细节的关注,并感谢您的宝贵意见。
真诚地
刘玉明
Reviewer 3 Report
Comments and Suggestions for Authors
The Authors present a study evaluating the effects of BI 1015550 on silica-induced silicosis and LPS-induced acute lung injury. The study is very well organized, evaluating the anti-fibrotic and anti-inflammatory effects of the tested molecule with different methodological approaches. However, some aspects of the manuscript would need to be improved in order for it to be published. Some specific suggestions follow:
- the resolution of figures in which some symbols, such as #, are not clearly visible should be improved;
- the manuscript requires careful revision for typos (see "silicosisis at line 31) and different font sizes;
- line 24: "LPS" should be written in full the first time it is mentioned;
- line 103: why is treatment with dexamethasone or BI 1015550 carried out before treatment with LPS? Please, clarify;
- line 143: what is the function of P-DMAB? Please add it;
- lines 235-239: this period is redundant with respect to the methods text. Additionally, I would recommend separating Figure 1A and 3A and citing them in the methods; even the captions are redundant;
- discussion: the differences in the effects of BI 1015550 compared to the positive controls used, i.e., nintedanib and dexamethasone, should also be discussed. Given the lower efficiency of BI 1015550 compared to dexamethasone, have levels greater than 0.75 mg/kg of BI 1015550 been tested? Are they tolerated?
Author Response
Dear Reviewers,
Thank you for your constructive feedback. We have carefully revised the manuscript based on the comments. Below is our point-by-point response.
Comments 1: The resolution of figures in which some symbols, such as #, are not clearly visible should be improved.
Response 1:We sincerely appreciate your insightful comments, which have helped us improve the quality of this manuscript. I'm sorry for the unclear symbols in the manuscript. We will correct the font size of some symbols in the article and improve the resolution to make the data more clearly visible. Your suggestion helps enhance the depth of our analysis, and we will make sure to address this in our revisions. Thank you for bringing these points to our attention.
Comments 2: The manuscript requires careful revision for typos (see "silicosisis at line 31) and different font sizes.
Response 2:Thank you for your review and feedback on our manuscript. We greatly appreciate that you can point this out, which will help to enhance the fluency of our manuscript. We apologize for the typographical errors and inconsistent font sizes in the article. We will review the entire text to avoid similar issues. At the same time, we will adjust the font size of the article and homogenize it. We appreciate your valuable input.
Comments 3: Line 24: "LPS" should be written in full the first time it is mentioned.
Response 3:Dear Reviewer, Thank you for your insightful comments. We thank you for bringing this issue to our attention, which will help to enhance the rigor of the manuscript. We will indicate the full name of LPS upon its first mention to avoid confusion and enhance the rigor of the article. This information will be clarified in the manuscript (Line 26).We appreciate your attention to detail and thank you for your valuable input.
Comments 4: Line 103: why is treatment with dexamethasone or BI 1015550 carried out before treatment with LPS? Please, clarify.
Response 4:Dear Reviewer, Thank you for your valuable feedback. The reason we choose to administer the drug before inducing acute lung injury in mice with LPS is that we have comprehensively considered the disease mechanism and pharmacodynamics, aiming to maximize the protective effect of the drug. LPS can rapidly trigger inflammatory responses in the body (such as the release of TNF-α, IL-6). If medication is given after modeling, it may not effectively block early inflammatory signals. From the perspective of pharmacodynamics, dexamethasone, as a glucocorticoid, needs to be administered in advance to maximize its anti-inflammatory effect. Furthermore, many studies reported have administered drugs before inducing acute lung injury with LPS to maximize the effect of the medication. For instance, Hu et al. administered Capsaicin for five consecutive days before LPS modeling and demonstrated that Capsaicin has a mitigating effect on acute lung injury in mice [1]. Hong et al. administered Hydnocarpin D for three consecutive days before LPS modeling and proved that Hydnocarpin D can inhibit inflammatory responses and improve LPS-induced acute lung injury in mice [2]. Therefore, considering all aspects comprehensively, we choose to administer the drug before inducing acute lung injury with LPS, to maximize the effect of the medication. We appreciate your attention to detail and thank you for your valuable input.
[1]Hu Q, Liu H, Wang R, Yao L, Chen S, Wang Y, Lv C. Capsaicin Attenuates LPS-Induced Acute Lung Injury by Inhibiting Inflammation and Autophagy Through Regulation of the TRPV1/AKT Pathway. J Inflamm Res. 2024 Jan 9;17:153-170. doi: 10.2147/JIR.S441141.
[2]Huanwu Hong, Siyue Lou, Fanli Zheng, Hang Gao, Nina Wang, Shasha Tian, Guozheng Huang, Huajun Zhao, Hydnocarpin D attenuates lipopolysaccharide-induced acute lung injury via MAPK/NF-κB and Keap1/Nrf2/HO-1 pathway, Phytomedicine, 2022, 101: 154143, doi: 10.1016/j.phymed.2022.154143.
Comments 5: Line 143: what is the function of P-DMAB? Please add it.
Response 5:Dear Reviewer,Thank you for your inquiry. We thank you for bringing this issue to our attention, which will help to enhance the rigor of the manuscript. P-DMAB (p-dimethylaminobenzaldehyde) acts as a chromogenic agent here. The sample undergoes acid hydrolysis to produce free hydroxyproline, which is oxidized by chloramine-T into pyrrole. The pyrrole can then condense with P-DMAB to form a red compound. The product has a characteristic absorption peak at 577 nm, and the hydroxyproline content can be quantitatively detected by changes in absorbance value. This information will be clarified in the manuscript (Lines 155-157). We appreciate your attention to detail and thank you for your valuable input.
Comments 6: Lines 235-239: this period is redundant with respect to the methods text. Additionally, I would recommend separating Figure 1A and 3A and citing them in the methods; even the captions are redundant.
Response 6:Dear Reviewer, Thank you for your valuable suggestions. We believe your suggestions are very reasonable and we have made appropriate modifications in the article. We have removed the synthetic route and placed it in the Supplemantal information. Furthermore, we have separated Figure 1A and Figure 3A, placing them in the method section and citing them there (Lines 116-124). We appreciate your attention to detail and thank you for your valuable input.
Comments 7: Discussion: the differences in the effects of BI 1015550 compared to the positive controls used, i.e., nintedanib and dexamethasone, should also be discussed. Given the lower efficiency of BI 1015550 compared to dexamethasone, have levels greater than 0.75 mg/kg of BI 1015550 been tested? Are they tolerated?
Response 7:Dear Reviewer,Thank you for your inquiry. We believe that your suggestions are very beneficial for increasing the scientific research depth of our article, and we have made timely changes. We supplement in the article the difference in treatment effects of BI 1015550 compared to Nintedanib and Dexamethasone and analyze the possible reasons. In our study, we have found that compared to Nintedanib, BI 1015550 significantly improved dynamic compliance (Cdyn) and expiratory airway resistance (RE) and reduced the expression levels of Fn and Col-1 mRNA in the lung tissues. Especially in terms of the IL-1β indicator, BI 1015550's inhibitory effect on IL-1β was slightly stronger than that of Nintedanib, whether in Bronchoalveolar lavage fluid (BALF) or in lung tissue. We believe that the anti-inflammatory and anti-fibrotic effects of BI 1015550 are slightly superior to those of Nintedanib. We believe this may be due to BI 1015550 simultaneously inhibiting the core pathways of inflammation and fibrosis (such as TGF-β, cAMP-PKA), and this dual inhibition demonstrates a stronger anti-inflammatory and anti-fibrotic efficacy[1]. Additionally, Nintedanib mainly targets fibroblasts and endothelial cells, with weaker regulation during the inflammatory phase. BI 1015550, on the other hand, can act on both inflammatory cells (macrophages, neutrophils) and structural cells (fibroblasts, epithelial cells), blocking multiple stages of the fibrotic process[2]. Therefore, the anti-inflammatory and anti-fibrotic effects of BI 1015550 may be stronger than those of Nintedanib, and the specific mechanisms require further study.
Later, we have found that the anti-inflammatory effect of BI 1015550 is lower than that of Dexamethasone, we believe it may be because Dexamethasone, as a glucocorticoid, has a very broad range of effects, capable of influencing gene transcription levels and simultaneously inhibiting multiple inflammatory pathways and mediators. In contrast, BI 1015550, as a PDE4B inhibitor, has a relatively more limited mechanism of action. Since PDE4B is primarily found in inflammatory cells, although inhibiting PDE4B may lead to an increase in intracellular cAMP levels, thereby suppressing the activation of these cells and the release of inflammatory mediators, this effect may be limited to specific inflammatory cells and mediators, unlike glucocorticoids that affect a wide range of inflammatory pathways. Therefore, its anti-inflammatory effect is not as strong as that of Dexamethasone. However, considering safety, long-term use of Dexamethasone may lead to severe side effects (such as immunosuppression, metabolic disorders), thus BI 1015550 may serve as a milder anti-inflammatory drug that can be used for long-term treatment to inhibit disease progression. The specific mechanism of BI 1015550 in treating ALI requires further study. This information will be clarified in the manuscript (Lines 350-439).
Given that the therapeutic effect of BI 1015550 is lower than that of Dexamethasone, we intend to select a higher dose (2.5 mg/kg) of BI 1015550 for experiments in future studies to further explore its effect in treating acute lung injury. We appreciate your attention to detail and thank you for your valuable input.
[1]Yuming Liu, Zhigang Liu, Xiaohe Li,Wenqi Li, et al. Nerandomilast Improves Bleomycin-Induced Systemic Sclerosis-Associated Interstitial Lung Disease in Mice by Regulating the TGF-β1 Pathway. Inflammation, 2024. doi: 10.1007/s10753-024-02153-9.
[2]Herrmann FE, Hesslinger C, Wollin L, et al. BI 1015550 is a PDE4B Inhibitor and a Clinical Drug Candidate for the Oral Treatment of Idiopathic Pulmonary Fibrosis. Front Pharmacol, 2022, 13: 838449. doi: 10.3389/fphar.2023.1219760.
Sincerely,
Yuming Liu
Round 2
Reviewer 3 Report
Comments and Suggestions for Authors
The Authors have addressed all the Reviewer's comments/suggestions.